# Modelling Lithium-Ion Battery Ageing in Electric Vehicle Applications—Calendar and Cycling Ageing Combination Effects

**Eduardo Redondo-Iglesias** [1,2,*] **, Pascal Venet** [2,3] **and Serge Pelissier** [1,2]

1   AME-Eco7, Univ Gustave Eiffel, IFSTTAR, Univ Lyon, 69500 Bron, France; serge.pelissier@univ-eiffel.fr
2   ERC GEST (IFSTTAR/Ampère Joint Research Team for Energy Management and Storage for Transport), 69500 Bron, France; pascal.venet@univ-lyon1.fr
3   Univ Lyon, Université Claude Bernard Lyon 1, École Centrale de Lyon, INSA Lyon, CNRS, Ampère, F-69100 Villeurbanne, France
*   Correspondence: eduardo.redondo@univ-eiffel.fr

**Abstract:** Battery ageing is an important issue in e-mobility applications. The performance degradation of lithium-ion batteries has a strong influence on electric vehicles' range and cost. Modelling capacity fade of lithium-ion batteries is not simple: many ageing mechanisms can exist and interact. Because calendar and cycling ageings are not additive, a major challenge is to model battery ageing in applications where the combination of cycling and rest periods are variable as, for example, in the electric vehicle application. In this work, an original approach to capacity fade modelling based on the formulation of reaction rate of a two-step reaction is proposed. A simple but effective model is obtained: based on only two differential equations and seven parameters, it can reproduce the capacity evolution of lithium-ion cells subjected to cycling profiles similar to those found in electric vehicle applications.

**Keywords:** battery ageing model; lithium-ion battery; electric vehicle

## 1. Introduction

Lithium-ion batteries constitute the most reliable energy storage technology for electric applications where energy density is critical. This is the case of electric vehicles, smart devices and portable power tools. Lithium-ion is nowadays a mature technology: cost and lifetime were improved in a very sensitive way in the last decades.

However, studying the ageing of batteries is still necessary because the degradation of their features largely determines the cost, the performances and the environmental impact of electric vehicles, particularly of full electric vehicles. In this type of studies, battery ageing is typically classified in two types: calendar and cycling ageing. Calendar ageing occurs when a battery is at rest condition; this is when no current flows through the battery whereas cycling ageing occurs when the battery is charged or discharged.

Given that battery degradation occurs in a different way if the battery is in rest condition or if a current flows through, a major challenge is to determine how calendar and cycling ageing effects combine together. Electric cars spend most of the time (95% or more) parked and current rates of the battery are relatively low when they are used. In these applications the average current rates are frequently about $C/5$ to $C/2$ with peak values at about $3C$. Except for fast charge, the main ageing mechanism in this application is considered to be formation and growth of the Solid Electrolyte Interface (SEI) [1,2].

The chosen method in this work is divided in two distinct phases, namely characterisation and modelling: The characterisation phase is based on accelerated ageing testing of battery cells, the main results of this phase were reported in Reference [3]. In this paper we are focusing in second phase: battery ageing modelling.

Battery ageing is modelled using the results obtained in the first phase. In our approach, we aim to establish ageing laws, that is to find the relations between ageing test conditions and performances decay and to quantify them. These ageing laws can be determined from test results and then generalised to predict the performance degradation of a battery subjected to different use conditions.

This paper is organised as follows: Main battery ageing mechanisms and modelling approaches are explained in Section 2. The experimental setup and results are reported in Section 3. In Section 4, a calendar ageing model is developed and its parameters are identified to fit experimental results. Then, a combined ageing model (cycling + calendar) is developed in Section 5. Finally, results are discussed and conclusions are drawn in Sections 6 and 7.

The obtained ageing model is able to reproduce the non linear behaviour of different combinations of calendar and cycling periods. Other reliability approaches (for example event-oriented modelling) could not explain these non linearities in a simple manner. Our model is simple but effective: it lies in a low number of equations (2 differential equations) and 7 parameters and enables to simulate the capacity fade of a battery cell subjected to ageing conditions combining cycling and rest periods. This model can be used for example to optimise the design and use of the battery in a vehicle by minimising both energy consumption and battery degradation.

## 2. Battery Ageing

### 2.1. Main Ageing Mechanisms in Lithium-Ion Batteries

Battery ageing relies on parasitic physico-chemical reactions occurring between the different components in a battery cell: electrodes, electrolyte, current collectors, additives. These mechanisms degrade storable energy (capacity) and maximum power (impedance) of the battery. Each ageing mechanism may depend of temperature ($T$), state of charge ($SoC$) and current $I$. Some mechanisms are present in calendar ageing ($I = 0$), while others are activated by cycling ($I \neq 0$). In the literature, many extensive bibliographic papers can be found, for example References [1,2,4,5].

The degradation of lithium-ion batteries is assumed to depend on three fundamental factors: $T$, $I$, and $SoC$. It should be underlined that not only the instantaneous value of these three factors, but also their temporal variations can impact the battery life. For example, in ageing tests campaigns in References [6,7] batteries were cycled at different levels of $SoC$ and different amplitudes ($DoD$). In these two works an important influence of the cycling amplitude was found. As pointed out by Reference [5], $SoC$ (or $DoD$) influence on ageing is not simple to analyse, because in some situations low $SoC$ levels (high $DoD$) can be beneficial while it could be harmful in other cases. Finally, a very non linear $SoC$ dependence of cycling ageing was identified experimentally by Reference [8]. In that work, five cycling tests of the same $SoC$ amplitude (20%) at different average $SoC$ levels were carried out. Cycling at very low and at very high levels of $SoC$ (0% to 20% and 80% to 100%) caused respectively the slowest and the fastest degradations, but no big difference was found between intermediary $SoC$ levels (20% to 40%, 40% to 60% and 60% to 80%).

In modern lithium-ion batteries, the main calendar ageing mechanism is the growth of the Solid Electrolyte Interface (SEI) layer on the negative (graphite) electrode [9,10]. It consists in an electrolyte reduction by the lithium that should be inserted into the graphite electrode. The composition of the SEI layer is very complex because it depends of the composition of the electrolyte solvents and electrode additives. In fact, the SEI layer is a multiple reaction mechanism [11].

SEI formation is accelerated at high levels of temperature ($T$) and State of Charge ($SoC$) [1]. Depending on the positive electrode composition, SEI growth may be accelerated by other mechanisms, particularly manganese or iron dissolution and migration at high levels of $T$ and $SoC$ [1,12].

The most representative cycling ageing mechanism is the lithium plating on the negative (graphite) electrode. This mechanism consists in a diffusion limitation of lithium insertion when the battery is charged at high current rates or low temperatures. In this conditions, lithium may be deposed on the negative electrode instead to be inserted into graphite [13,14].

Other cycling ageing mechanisms in lithium-ion batteries are, for example, particle cracking and collector corrosion. However, this type of mechanisms occurs mostly in extreme use conditions at very high current rates or very deep discharges, not in normal use conditions [1].

### 2.2. Modelling Approaches

A classification of ageing modelling approaches was made in Reference [15]. In that work, three approaches were considered—physico-chemical, weighted Ah and event-oriented ageing modelling.

In the physico-chemical approach, internal state of each battery component (electrodes, current collectors, electrolyte) is modelled. This approach can be interesting to better understand ageing mechanisms and interactions between battery components, but it needs a heavy experimental setup. Moreover, this type of models are usually very complex with a high number of multi-variable differential equations and a high number of model parameters to determine.

The weighted Ah model is an empirical approach where battery performances are modelled as a function of Ah-throughput. Here, the main assumption is that ageing is directly related to quantity of charge (Ah-throughput) delivered by the battery. Depending on use conditions (temperature, *SoC*, etc.), ageing will be faster or slower for the same Ah-throughput, and this model attempts to take into account these differences by weighting discharged Ah according the stress factors.

The event-oriented model is another empirical approach where battery use is divided in phases (events). The main assumption here is to suppose that the performance degradation produced by an event is independent of past events.

With the weighted Ah or the event-oriented approaches, it is difficult to consider interactions between use phases, as for example, the interactions between calendar and cycling phases that have been found in Reference [3].

Many preceding works focused either in modelling calendar ageing [16–18] or cycling ageing [19,20]. The problem is that, in real life, batteries are sometimes at rest (calendar ageing), sometimes in use (cycling ageing). To cope with this, some works (e.g., Reference [21]) consisted in decomposing battery use in rest phases and cycling phases: when battery is at rest the used ageing model is a calendar one and a cycling ageing model is used during use phases. Other authors considered that calendar and cycling ageing are additive [6,7]. The problem with these approaches is to deal interactions existing between calendar and cycling.

In this paper we are focusing in the combination of calendar and cycling ageing. This is a continuation of a preceding work reported in Reference [3], where it has been shown that calendar and cycling ageing are not additive: we cannot simply add degradations produced at each use phase. From the results of that work, a strong interaction between calendar and cycling exists, even at low current rates.

## 3. Experiments

### 3.1. Experimental Setup

The ageing test campaign was designed to put into evidence the existing interactions between cycling phases and rest periods on ageing. Accelerated ageing tests were conducted on 36 lithium-ion cells (KOKAM, nominal capacity 0.35 Ah, NMC/C) at 60 °C. For practical reasons we have chosen smaller cells (0.35 Ah) for the test campaign. Notice that the same cell technology exists in a bigger format for EV applications (e.g., 12 Ah) with a similar composition. Some cells were tested in calendar ageing and others in cycling ageing. Calendar ageing consisted in leaving the cells at rest condition (disconnected) at 5 different *SoC* levels: 100, 90, 80, 70 and 50%. Cycling ageing consisted in periodically

charge and discharge the cells following 7 different profiles. In this work we will focus on the results of profiles shown in Figure 1, for further details about these experiments, a full description and analysis of ageing test results can be found in Reference [3]. Profile a represents a daily use of an electric vehicle, for instance: home to work, then work to home with a full charge at the end of the day. Profile b is a variation of profile a without the return trip. Profiles c and d are like profile b (same *SoC* levels), but with higher frequencies. Finally, profiles e and f were designed to investigate the influence of this type of cycling at lower *SoC* levels. Three cells were tested at each ageing condition to confirm the results repeatability.

Cycling ageing is performed at very low current rates (*C*/2 in discharge, *C*/5 in charge) compared to maximum allowed rates indicated by the battery manufacturer (20*C* in discharge, 2*C* in charge). These current levels are representative of those in electric vehicle applications. Moreover, as in electric vehicle application, cells are most of time at rest condition.

Cells' performances were periodically measured by the means of Reference Performance Tests (RPT). The RPT consisted in measuring the self-discharge and the cell capacity at two current rates (1*C* and *C*/10).

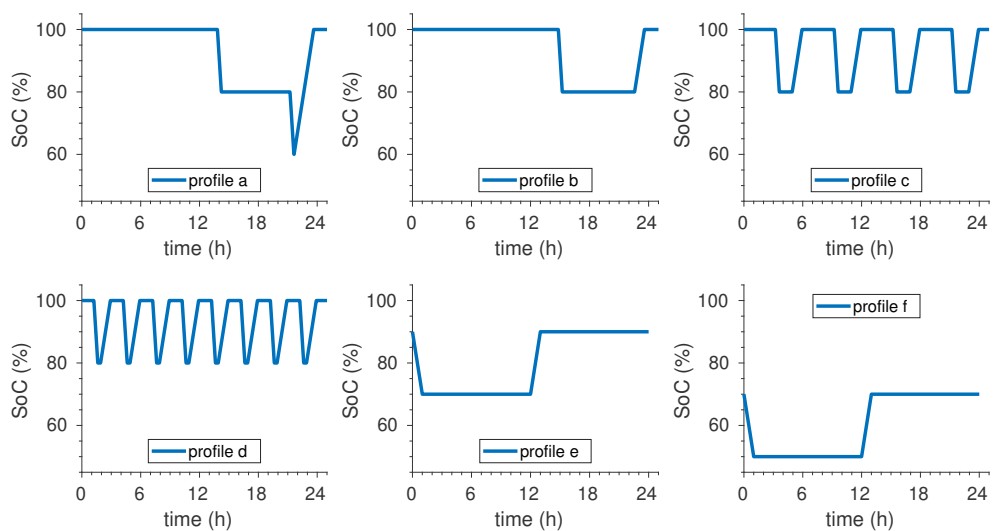

**Figure 1.** Cycling profiles.

### 3.2. Experimental Results

Once data are collected, they must be analysed to determine if differences in degradation rates exist between pure calendar ageing and combined (cycling/calendar) ageing. In Reference [3] an analysis of degradation rates has been made showing that an interaction exists even at low current rates (typically those of batteries in electric vehicle applications). It was found that degradation can be much faster when cycling phases are combined to calendar ageing periods respect to the case of pure calendar ageing.

Figure 2 shows the results (capacity fade) of ageing tests. In this figure we can see that calendar ageing is very sensible at higher *SoC* levels (90%, 100%) and it is nearly the same for lower levels (80%, 70%, 50%).

Concerning the cells subjected to cycling profiles, they spent most of time at rest. For example, for profile b, the cells are most of time at rest, either at 100% or at 80% *SoC*. If only calendar ageing existed in these tests, their capacity fade would be between that of calendar ageing at *SoC* 100% and that of *SoC* 80%. However, the degradation of these cells is very fast, at a similar rate than calendar ageing at *SoC* 100%, showing a strong influence of cycling phases on ageing even with low current rate.

On the contrary, for cells subjected to profiles e and f cycling influence is less important, showing that cycling influence can be very non linear.

A result that may seem surprising appears when comparing the capacity fade evolution of cells subjected to profiles a and b. These two profiles are identical, except for the second partial discharge of the profile a (Figure 1). All ageing factors ($T$, $I$, average $SoC$) are similar between these two profiles, except for the depth of discharge: $DoD$ in profile a is twice compared to $DoD$ in profile b. However, the cells subjected to profiles a or b evolved in the same way despite the fact that the $DoD$ is very different (Figure 2).

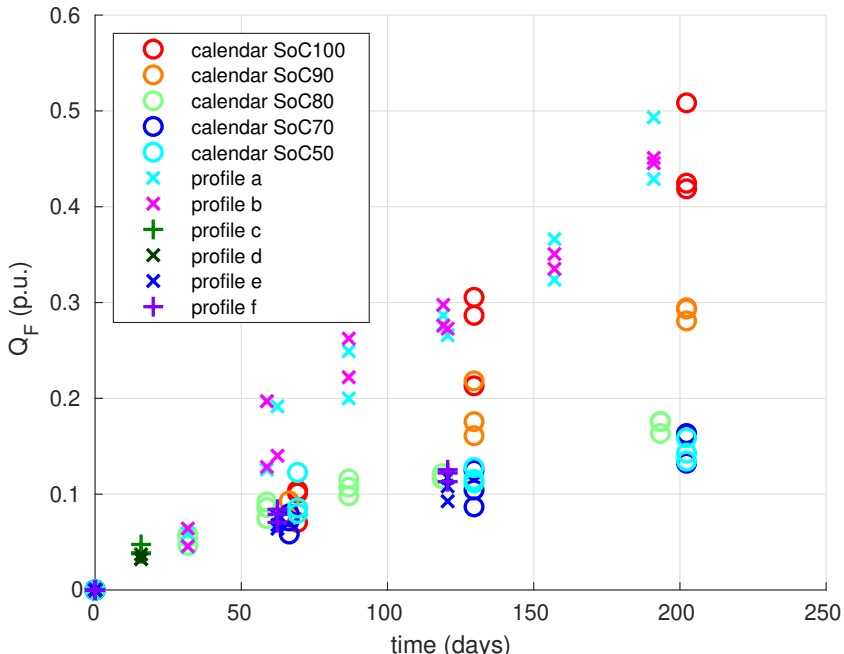

**Figure 2.** Capacity fade under different calendar/cycling ageing conditions. Relative values of capacity fade are expressed in *p.u.* which means "per unit".

Since in this work we are focusing on ageing modelling, the results are summarised. For further details about these experiments, a full description and analysis of ageing test results can be found in Reference [3].

## 4. Calendar Ageing Model

### 4.1. Model Formulation

In this work we are focusing on the SEI growth mechanism. As explained above, SEI layer composition is not simple and many reactions occur at the same time to form this layer. Nevertheless, in some cases several parallel multi-step reactions may be merged as one unique reaction (Equation (1)). Since capacity is proportional to the quantity of usable lithium, capacity fade is proportional to the quantity of immobilised lithium ions by each mechanism. Thus, the rate of this equivalent single-step reaction, $k_{SEI}$, is proportional to the capacity fade rate $\left(\frac{dQ_F}{dt}\right)$ or acceleration coefficient, $C_a$ (Equation (2)).

$$\text{Reactants} + \text{Li}^+ \xrightarrow{\ k_{SEI}\ } \text{SEI} \tag{1}$$

$$Q \propto \text{quantity of usable Lithium} \Leftrightarrow k_{SEI} \propto \frac{dQ_F}{dt} = C_a. \tag{2}$$

In reliability assessment, acceleration models are developed to predict the time-to-fail ($t_f$): the time from Beginning-of-Life (*BoL*) to End-of-Life (*EoL*) of a system or a component. Some of these models are based on empirical models like Arrhenius and Eyring laws. The Arrhenius law was originally used to model the dependence of a reaction rate with temperature. Later, Eyring law was developed to expand the Arrhenius law to other stress factors [22].

The general form of the Eyring law for a stress factor $S_1$ is given in Equation (3). *A* is the pre-exponential factor, *n* is the temperature exponent, $E_a$ is the activation energy, *k* is the Boltzmann constant, $B_1$ the direct influence factor of $S_1$ and $C_1$ the interaction factor between $S_1$ and *T*. This law was used in preceding works to model capacity fade [23,24] as a function of temperature and *SoC* ($S_1 = SoC$).

$$C_a(T, S_1) = A \cdot T^n \cdot \exp\left[-E_a/(k \cdot T) + B_1 \cdot S_1 + (C_1 \cdot S_1)/(k \cdot T)\right] \tag{3}$$

An interesting feature of Eyring law is that it is easily expandable to other stress factors, for example, a second stress factor can be added to Equation (3) by adding a term $B_2 \cdot S_2 + (C_2 \cdot S_2)/(k \cdot T)$.

Due to the non-linearity of battery behaviour respect to *SoC*, sometimes it is convenient to use another stress factor formulation. For example, in Equation (4), the *SoC* ageing dependence is expanded to several stress factors ($S_i = f_i(SoC)$). This is equivalent to consider two or more stress factors depending on the *SoC* ($S_1 = f_1(SoC)$, $S_2 = f_2(SoC)$, ...):

$$C_a(T, SoC) = A \cdot T^n \cdot \exp\left[-E_a/(k \cdot T) + B_1 \cdot f_1(SoC) + B_2 \cdot f_2(SoC) + \ldots\right]. \tag{4}$$

### 4.2. Parameter Identification

In this work, we are focusing in the *SoC* behaviour of ageing: experiments were performed on NMC/C cells at one single temperature (60 °C). For this reason, there is no way to calculate the parameters defining the temperature behaviour ($n$, $E_a$, $C_1$, $C_2$,...), so Equation (4) becomes Equation (5) and the influence of these parameters is gathered in $A'$:

$$C_a(SoC) = A' \cdot \exp\left[B_1 \cdot f_1(SoC) + B_2 \cdot f_2(SoC) + \ldots\right]. \tag{5}$$

A parameter identification is performed to fit our model ($C_a$) to the results of calendar ageing tests (Figure 2). This task has already been reported in References [23,24]. For the purpose of this work, we are not considering *SoC* drift influence in the identification process. In fact, a sensible *SoC* drift influence was found in LFP/C cells [23] but not in NMC/C cells which are studied here [24]. Another important assumption is to consider that $C_a$ is independent of the State of Health (*SoH*), meaning that at each *SoC* level a value of $C_a$ can be found independently of the current capacity loss. With these assumptions, at each calendar ageing condition capacity fade can be approximated to a straight line:

$$Q_F(t) = C_a \cdot t \tag{6}$$

The first step is to fit the capacity fade measurements of each tested cell (Figure 2, only calendar results) by performing linear regressions that is, for each cell *j*, $C_{a,j}$ is obtained as the slope of $Q_F(t)$.

The second step consist in gathering the obtained values of $C_{a,j}$ to fit them to an Eyring law defined in advance. Equation (5) can be transformed by logarithms to a multi-linear expression:

$$\ln(C_a(SoC)) = \ln(A') + B_1 \cdot f_1(SoC) + B_2 \cdot f_2(SoC) + \ldots \tag{7}$$

$$Z = \alpha + \beta \cdot X + \gamma \cdot Y + \ldots \tag{8}$$

Equation (7) is equivalent to a multi-linear equation (Equation (8)) where $Z = \ln(C_a(SoC))$, $X = f_1(SoC_j)$, $Y = f_2(SoC_j)$ and $\alpha = \ln(A')$, $\beta = B_1$ and $\gamma = B_2$. A multi-linear regression allows to identify the parameter values ($\alpha$, $\beta$, $\gamma$, …) from a set of value tuples ($X_j = f_1(SoC_j)$, $Y_j = f_2(SoC_j)$, $Z_j = \widehat{C}_{a,j}$). As mentioned above, three cells for each *SoC* level (50%, 70%, 80%, 90% and 100%) were tested, then there are fifteen tuples ($X_j$, $Y_j$, $Z_j$).

Finally, the Eyring parameters are respectively $A' = \exp(\alpha)$, $B_1 = \beta$ and $B_2 = \gamma$.

Figure 3 shows the natural logarithm of obtained values of $C_{a,j}$ versus *SoC* (blue circles). These values were obtained with linear regressions of $Q_F$, versus time (by Equation (6)). Notice that *SoC* of each cell drifts from the initial values due to both self-discharge and capacity fade (reversible and irreversible losses), this effect has been reported and explained in Reference [25]. For example, SoC50 cells were in fact at an average *SoC* between 40 and 50%. The results of the second step (i.e., multilinear regression of $C_a$ to obtain $A$ and $B$, Equation (7)) are also shown in Figure 3 for three different models (respectively Equations (9)–(11)):

$$C_a(SoC) = A' \cdot e^{(B \cdot SoC)} \tag{9}$$

$$C_a(SoC) = A' \cdot e^{(B \cdot SoC^z)} \tag{10}$$

$$C_a(SoC) = A' \cdot e^{(B \cdot f_{re}(SoC))} \tag{11}$$

Model 1 (Equation (9)), is the typical Eyring law considering directly *SoC* as stress factor ($f_1(SoC) = SoC$). The logarithmic plot versus *SoC* shows this model as a linear function. This model fits very well for every *SoC* from 70% to 100%. However, $C_a$ stop decreasing under 70% *SoC*. The model performs well for every *SoC* except for 50%: at this *SoC*, $Q_F$ is underestimated.

Choosing other functions may improve the fitting results. For example, in model 2 (Equation (10)) a power function of *SoC* is used: $f_1(SoC) = SoC^z$. For simplicity reasons, it has been decided to fix the value of $z$ instead to include it in the identification process. So the identification algorithm outputs the values of $A'$, $B$ for a each chosen value of $z$. Different values of $z$ from 1 to 10 were tested and $z = 5$ provided the lowest errors compared to RPT measurements (Table 1). This model provides more satisfactory results than model 1, especially for 50%. Nevertheless, the capacity fade is now slightly underestimated at $SoC = 90\%$.

Other functions were tested by using different combinations of $\sqrt{SoC}$, $SoC$, $SoC^2$, $SoC^3$. The results are even better with absolute mean errors under 4% of initial capacity (Table 1). But these models have the inconvenient of complexity, because they need more than two parameters.

Finally, an "*exponential ramp function*" ($f_{re}(SoC)$) was tested in Equation (11)). $f_{re}(SoC)$ is defined in Equation (12) and the returned values by this function are quite similar to those returned by a ramp function ($f_r(SoC)$, Equation (13)). For values of *SoC* lower than $a$, $f_{re}(SoC) \simeq f_r(SoC) = a$; for values greater than $a$, $f_{re}(SoC) \simeq f_r(SoC) = SoC$. The advantage of $f_{re}(SoC)$ over $f_r(SoC)$ is the derivability.

As illustrated by Figure 3, model 3 gives the best result. While models 1 and 2 underestimate $Q_F$ for $SoC = 50\%$ and 90% respectively, model 3 does not. The accuracy of model 3 was judged to be satisfactory with errors in the order of magnitude of the dispersion due to differences in cell manufacturing. For this reason, for the following sections (combined ageing modelling) we have adopted model 3 for $C_a$.

$$f_{re}(SoC) = a + \frac{(SoC - a)}{(1 + e^{-b(SoC - a)})} \tag{12}$$

$$f_r(SoC) = \begin{cases} a, & SoC \leq a \\ SoC, & SoC \geq a \end{cases} \tag{13}$$

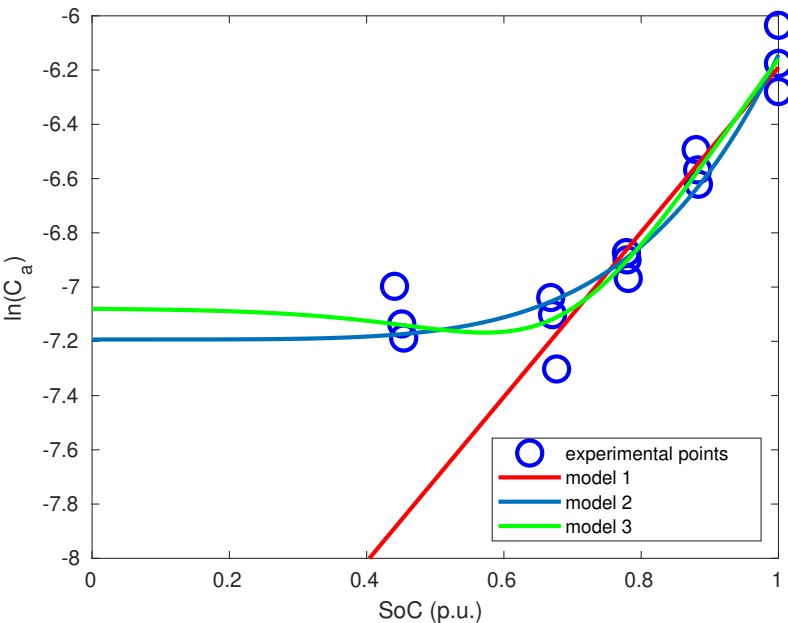

**Figure 3.** Calendar ageing modelling results: natural logarithm of acceleration coefficient ($C_a$) versus *SoC*. The blue circles are the experimental points, the lines are the result of model 1, 2 and 3 (Equations (9), (10) and (11) respectively). For model 2: $z = 5$. For model 3: $a = 0.7, b = 10$.

**Table 1.** Calendar ageing model results: absolute mean errors and maximum errors of $C_a$ for different stress formulations.

| $C_a$ | abs. Mean Error /Max. Error (%) | Identified Parameters | Fixed Parameters | Comments |
|---|---|---|---|---|
| | 7.1/22.5 | | $z = 1$ | model 1 (Equation (9)) |
| | 6.2/22.1 | | $z = 2$ | |
| | 5.0/17.1 | | $z = 3$ | |
| | 4.1/13.1 | | $z = 4$ | |
| $A' \cdot \exp\left(B \cdot SoC^z\right)$ | 4.0/11.4 | $A', B$ | $z = 5$ | model 2 (Equation (10)) |
| | 4.4/13.2 | | $z = 6$ | |
| | 4.9 /14.4 | | $z = 7$ | |
| | 5.4/15.0 | | $z = 8$ | |
| | 5.8/15.2 | | $z = 9$ | |
| | 6.1/15.6 | | $z = 10$ | |
| $A' \cdot \exp\left(B_1 \cdot \sqrt{SoC} + B_2 \cdot SoC\right)$ | 3.7/11.4 | $A', B_1, B_2$ | | |
| $A' \cdot \exp\left(B_1 \cdot SoC + B_2 \cdot SoC^2\right)$ | 3.8/12.2 | $A', B_1, B_2$ | | |
| $A' \cdot \exp\left(B_1 \cdot SoC + B_2 \cdot SoC^2 + B_3 \cdot SoC^3\right)$ | 3.5/11.9 | $A', B_1, B_2, B_3$ | | |
| $A' \cdot \exp\left(B \cdot f_{re}(SoC)\right)$ | 3.5/11.7 | $A', B$ | $a = 0.7, b = 10$ | model 3 (Equation (11)) |

## 5. Combined Ageing Model

### 5.1. Model Formulation

As explained above, the main hypothesis is to consider that ageing is mainly calendar and due to SEI growth. The effect of cycling on ageing is an acceleration of calendar ageing: this means that calendar ageing rate is modified by charges (and discharges).

The precise side reactions behind such ageing behaviour will remain unknown because a lack of analysis equipments. A possible explanation would be the following: cycling induces volume changes on the electrode; these dilatations make the SEI layer to be more propitious to further electrolyte reduction and thus to SEI growth. This should be validated by physico-chemical analyses.

To model this behaviour, we consider that SEI formation is a two-step reaction (Equation (14)). This reaction is inspired by Reference [26]. In that article, SEI formation is suggested to be formed by a two-step reaction. First step is a reversible reaction consisting in the formation of a complex between electrolyte solvent and lithium ions. The second one is the irreversible transformation of the previously formed complex to form SEI layer.

Consider the following multi-step reaction where X is irreversibly transformed into Z through the intermediary substance Y:

$$X \xrightleftharpoons[k_{YX}]{k_{XY}} Y \xrightarrow{k_Z} Z \tag{14}$$

If reactions are zero order respect to X and first order respect to Y, the transformation rate is defined by the following ordinary differential equations (ODE) system, where $[X]$, $[Y]$ and $[Z]$ are the concentrations of X, Y and Z:

$$-\frac{d[X]}{dt} = k_{XY} - k_{YX}[Y] \tag{15}$$

$$\frac{d[Y]}{dt} = k_{XY} - k_{YX}[Y] - k_Z[Y] \tag{16}$$

$$\frac{d[Z]}{dt} = k_Z[Y] \tag{17}$$

Reaction (14) can represent the lithium loss which is directly related to capacity fade, then let assume the following equivalences: $Q \equiv [X]$, $Q_{F,rev} \equiv [Y]$ and $Q_F \equiv [Z]$, where $Q$, $Q_{F,rev}$ and $Q_F$ are respectively the *capacity*, *reversible capacity fade* and *capacity fade*.

The operation of the proposed model is summarised by Figure 4: $Q$ is first transformed in $Q_{F,rev}$ depending of the values of $Q_{F,rev}^{eq}$ and $k_s \cdot I$. After this, a part of $Q_{F,rev}$ is reversibly transformed to $Q$ and the rest is irreversibly transformed to $Q_F$ ($k_{rev} + k_{irr} = 1$).

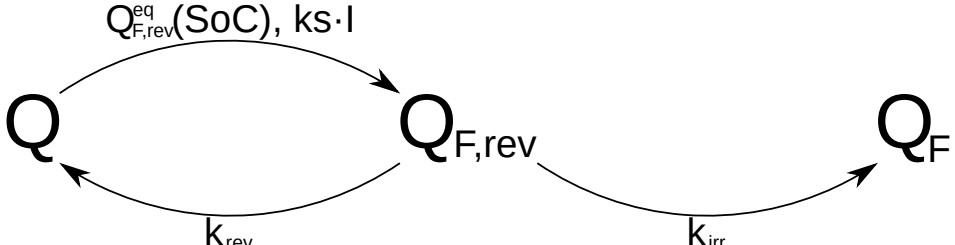

**Figure 4.** Combined model diagram.

Equations (15)–(17) can be rewritten to put into evidence the influence of the model paramaters on the dynamic evolutions of $Q$, $Q_{F,rev}$ and $Q_F$. To introduce the cycling effect into the model, we propose to add a term that modifies the *reversible capacity fade* rate ($dQ_{F,rev}/dt$). This term is proportional to the current: $k_s \cdot I$. The new ODE system is composed of Equations (18)–(20):

$$\frac{dQ}{dt} = -\lambda \cdot (Q_{F,rev}^{eq} - k_{rev} \cdot Q_{F,rev}) - k_s \cdot I \tag{18}$$

$$\frac{dQ_{F,rev}}{dt} = \lambda \cdot (Q_{F,rev}^{eq} - Q_{F,rev}) + k_s \cdot I \tag{19}$$

$$\frac{dQ_F}{dt} = \lambda \cdot k_{irr} \cdot Q_{F,rev} \tag{20}$$

The relations between the parameters of both ODE systems are:

$$\lambda = k_{YX} + k_Z \tag{21}$$

$$Q_{F,rev}^{eq} = \frac{k_{XY}}{\lambda} \tag{22}$$

$$k_{irr} = \frac{k_Z}{\lambda} \tag{23}$$

$$k_{rev} = \frac{k_{YX}}{\lambda} = 1 - k_{irr} \tag{24}$$

The meaning of the new parameter set is the following: $k_{rev}$ and $k_{irr}$ define the distribution of $Q_{F,rev}$ decomposition into reversible and irreversible parts ($k_{rev} + k_{irr} = 1$). $\lambda$ defines the response speed of the system. In other words, $1/\lambda$ is the time constant of the system. Finally, $Q_{F,rev}^{eq}$ is the equilibrium level of $Q_{F,rev}$ when the system is not forced ($I = 0$).

Since $Q$, $Q_{F,rev}$ and $Q_F$ are capacities (quantities of charge), they are directly related to mass (quantity of lithium) and they are subjected to mass constraints:

(i) Mass is non-negative, then capacities must be non-negative:

$$Q(t),\ Q_{F,rev}(t),\ Q_F(t) \geq 0 \tag{25}$$

(ii) Conservation of mass: the sum of capacities is constant and equal to initial capacity $Q^0$:

$$Q(t) + Q_{F,rev}(t) + Q_F(t) = Q^0. \tag{26}$$

Equation (26) allows to reduce the number of ODE's of the system given by Equations (18) to (20). Thereafter, we will focus on Equations (19) and (20) enabling to calculate $Q_{F,rev}$ and $Q_F$. $Q$ can be calculated afterwards by using Equation (26). Finally, the initial conditions are typically the following: $Q^0 = Q_{nom}$ and $Q_{F,rev}^0 = Q_F^0 = 0$.

In conclusion, this model is defined by 2 independent ODE's (Equations (19) and (20)), 2 constraint equations (Equations (25) and (26)), 4 independent (Note that $k_{rev}$ is directly related to $k_{irr}$, Equation (24)) parameters ($\lambda$, $Q_{F,rev}^{eq}$, $k_{irr}$ and $k_s$) and a set of initial conditions ($Q^0$, $Q_{F,rev}^0$ and $Q_F^0$).

In order to facilitate comparisons between different battery sizes, capacities are expressed in the *p.u.* system (capacity relative to nominal capacity of the battery, $Q_{nom}$). Thus, every capacity ($Q$, $Q_{F,rev}$ and $Q_F$) can be valued between 0 and 1 p.u.

The chosen unit of time is the day, then capacity rates $dQ/dt$, $dQ_{F,rev}/dt$ and $dQ_F/dt$ are expressed in p.u./day. In order to homogenise the units of the preceding equations, current ($I$), which is typically expressed in *C*-rate, will be expressed also in p.u./day. In fact, the *C*-rate system are units of p.u./hour: $1C$ is the current rate discharging (or charging) the battery in 1 hour. The conversion factor between *C*-rate (p.u./hour) and p.u./day is 24. For example, $1C$ is equivalent to 24 p.u./day and $C/24$ is equivalent to 1 p.u./day.

An important property of this model is that, in calendar conditions, it is equivalent to the calendar model proposed in Section 4. When no current flows through the battery $I = 0$, the term $k_s \cdot I$ of Equation (19) is 0. In these conditions, $Q_{F,rev}$ evolution is that of a first order system, that is, $Q_{F,rev}$ will converge to $Q_{F,rev}^{eq}$ from its initial value $Q_{F,rev}^0$. Then, after a certain time the equilibrium is found:

$$\frac{dQ_{F,rev}}{dt} = 0 \Leftrightarrow Q_{F,rev} = Q_{F,rev}^{eq} \tag{27}$$

$$\frac{dQ_F}{dt} = \lambda \cdot k_{irr} \cdot Q_{F,rev}^{eq} = C_a(SoC). \tag{28}$$

By using Equations (11) and (28), $Q_{F,rev}^{eq}$ can be expressed as a function of calendar ageing parameters obtained in the preceding Section ($A'$ and $B$) and combined ageing parameters ($\lambda$ and $k_{irr}$):

$$Q_{F,rev}^{eq}(SoC) = \frac{A' \cdot e^{(Bf_{re}(SoC))}}{\lambda \cdot k_{irr}}. \tag{29}$$

### 5.2. Parameter Identification

The parameter identification consists in solving a non-linear problem: to find the minimum of $f(x)$, where $x$ is a parameter set ($\lambda, k_{irr}, Q_{F,rev}^{eq}, k_s$) and $f(x)$ is the simulation error respect to experimental results. This process is iterative: it means that for each parameter set $x = (\lambda, k_{irr}, Q_{F,rev}^{eq}, k_s)$, the error is evaluated and a new parameter set is established for next iteration until a satisfactory result is found (minimum of $f(x)$). In this work, $f(x)$ is defined as the mean value of the errors on each profile (index $i$ indicates the profile number):

$$f(x) = \overline{error(x, profile_i)}, \tag{30}$$

where the error is calculated as the mean absolute error of simulated capacity fade respect to measured capacity fade in RPT tests (index $j$ indicates the RPT number):

$$error(x, profile_i) = \overline{|Q_{F,sim}(j) - Q_{F,meas}(j)|}. \tag{31}$$

In Section 4, a calendar ageing model based on the acceleration coefficient, $C_a(SoC)$ was found. As explained above, in calendar conditions, the combined ageing model is equivalent to the calendar ageing one. With this relation (Equation (29)), the number of parameters to identify is decreased of one: at each iteration a three parameter set is established $x = (\lambda, k_{irr}, k_s)$ and $Q_{F,rev}^{eq}$ have to be obtained from $\lambda, k_{irr}$ and $C_a(SoC)$. The iterative process is summarised by the following steps:

(i)　establish a new parameter set: $x = (\lambda, k_{irr}, k_s)$
(ii)　calculate $Q_{F,rev}^{eq}(SoC)$ (Equation (29))
(iii)　run simulation on each cycling profiles to obtain $Q_{F,sim}$ (Equations (19) and (20))
(iv)　calculate the mean absolute error, $f(x)$ (Equation (30))
(v)　while a minimum is not found, go to step (i)

We have used Mathworks' *MATLAB* and *Optimisation Toolbox* to solve the parameter identification problem. The initial value of the parameter set, their constraints and the obtained parameter set are described in Table 2.

**Table 2.** Minimisation parameters.

| Parameter | Initial Value ($x_0$) | Upper Bound ($x_{max}$) | Lower Bound ($x_{min}$) | Obtained Value ($x_{end}$) |
|:---:|:---:|:---:|:---:|:---:|
| $\lambda$ | 10 | 20 | 0.1 | 7.41 |
| $k_{irr}$ | 0.1 | 0.5 | 0.0001 | 0.0547 |
| $k_s$ | 0.1 | 0.5 | 0.0001 | 0.0548 |

For the identification we have used profiles b to f. Profile a is the more complex one (see Figure 1) and is used as model validation profile. The obtained combined ageing model is summarised by Table 3. In this table, all the necessary equations and parameters are indicated.

The simulation results for all ageing conditions are shown in Figure 5. First, we can see that calendar ageing simulations (dashed lines) fit very well to measurements (circles).

For the cycling profiles, the results are not very good for profiles b and e: capacity fade is underestimated in profile b and it is overestimated in profile e. Nevertheless, the model reproduces quite well the capacity fade found by experiments for all other profiles (a, c, d, f), especially capacity fade for profile a that has not been used for identification (profile a).

**Table 3.** Summary of model parameters and equations.

**(a)** Model equations.

Main equations (ODE system):

| | |
|---|---|
| $\frac{dQ_{F,rev}(t)}{dt} = \lambda \cdot (Q^{eq}_{F,rev} - Q_{F,rev}(t)) + k_s \cdot I(t)$ | Equation (19) |
| $\frac{dQ_F(t)}{dt} = \lambda \cdot k_{irr} \cdot Q_{F,rev}(t)$ | Equation (20) |

Auxiliary equations:

| | |
|---|---|
| $SoC(t) = SoC^0 + \frac{\int I(t) \cdot dt}{Q^0}$ | *t* in *days* |
| | *I* in p.u./day |
| $f_{re}(SoC) = a + \frac{(SoC-a)}{(1+e^{-b(SoC-a)})}$ | Equation (12) |
| $C_a(SoC) = A' \cdot e^{(Bf_{re}(SoC))}$ | Equation (11) |
| $Q^{eq}_{F,rev}(SoC) = \frac{C_a(SoC)}{(\lambda \cdot k_{irr})}$ | Equation (29) |
| $Q(t) = Q^0 - Q_{F,rev}(t) - Q_F(t)$ | from Equation (26) |

**(b)** Model parameters.

| Parameter | Units | Value |
|---|---|---|
| $A'$ | p.u./day | $8.8765 \times 10^{-5}$ |
| $B$ | (no units) | 3.2162 |
| $a$ | (no units) | 0.7 |
| $b$ | (no units) | 10 |
| $\lambda$ | $day^{-1}$ | 7.41 |
| $k_{irr}$ | (no units) | 0.0547 |
| $k_s$ | (no units) | 0.0548 |

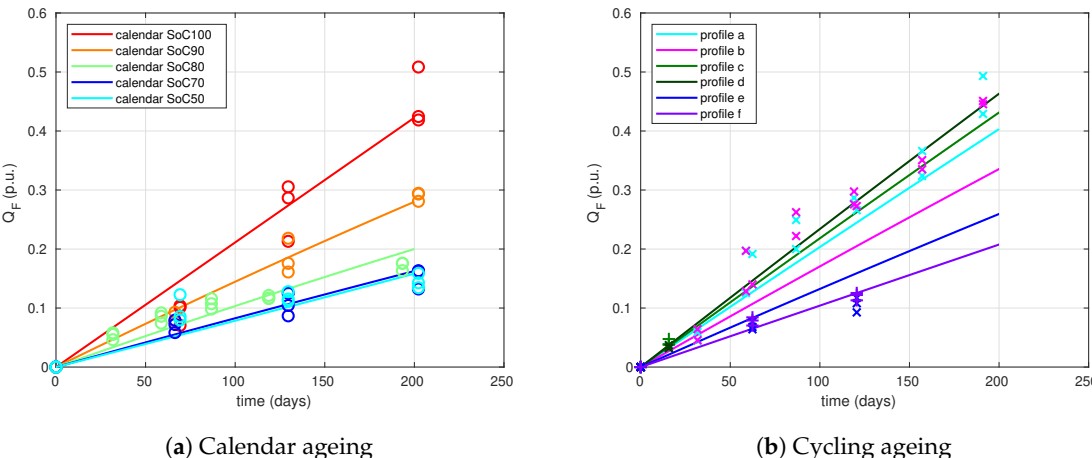

**(a)** Calendar ageing
**(b)** Cycling ageing

**Figure 5.** Capacity fade simulations compared to measurements under different ageing conditions. Simulations are plotted by using lines. Measurements are plotted with markers (the legend for these points is in Figure 2).

## 6. Discussion

In this section some application examples allow to illustrate the model behaviour particularities. The first example is composed of four different use profiles with the same amount of charge throughput and current rates (Figure 6). The first profile consists in a daily discharge of 20% *SoC* at *C*/2 followed by a 2 h rest and a full charge at *C*/2 (Figure 6a). In this use profile battery is at 100% *SoC* most of time: 21.2 h per day (blue line). The second profile consists in repeating once a week (on Monday)

seven times the following pattern: 20% *SoC* discharge at $C/2$, 2 hour rest, full charge at $C/2$ and 42 min rest. The battery is left at 100% *SoC* the rest of the week (red line). Rest times have been adjusted to make *SoC* levels (minimum, maximum and average) equal in profiles 1 and 2: the only difference is the distribution of charges and discharges within a week. The third and fourth profiles (yellow and violet lines respectively) are complementary two the preceding ones but the battery is kept at 80% *SoC* when it is not used. As for profiles 1 and 2, *SoC* levels of profile 3 are equal than those of profile 4.

The simulation results for one week (7 days) are illustrated by Figure 6b,c showing respectively $Q_{F,rev}$ and $Q_F$. As a consequence of ODE system formed by Equations (18)–(20), $Q_{F,rev}$ is first produced from $Q$. Then, $Q_{F,rev}$ is consumed: a part of $Q_{F,rev}$ is reversibly transformed to $Q$ and the other part is irreversibly lost ($Q_F$).

During each charge $Q_{F,rev}$ grows to about 0.016 *p.u.* Inversely, during each discharge, $Q_{F,rev}$ decreases rapidly and reaches the constraint value of 0 *p.u.* (Equation (25), non negativity of mass). During rest times, after each charge (discharge), $Q_{F,rev}$ will decrease (increase) to converge to a value depending of the *SoC* level: $Q_{F,rev}^{eq}(SoC)$. In other words, each use phase (charge or discharge) makes the cell move from an equilibrium ($Q_{F,rev} \neq Q_{F,rev}^{eq}$); when the battery is at rest, a new equilibrium is found between $Q_{F,rev}$ generation and consumption and it converges to a value depending of *SoC* ($Q_{F,rev} \simeq Q_{F,rev}^{eq}(SoC)$). In Figure 6b we can see that $Q_{F,rev}^{eq}$ is about 0.0025 *p.u.* for 100 % *SoC* (blue and red lines, during long rest periods), while it is about 0.0015 *p.u.* for 80% *SoC* (yellow and violet lines, during long rest periods).

The evolution of $Q_F$ is shown in Figure 6c. As we can see, after each charge there is an acceleration of $Q_F$ and inversely, after each discharge, $Q_F$ evolution decelerates. This behaviour is explained by Equation (20): $Q_F$ is proportional to the integral of $Q_{F,rev}$. The same pattern will be repeated during ten weeks as we can see in Figure 6d.

An important consequence of this model appears by comparing $Q_F$ of profiles 1 and 2 (Figure 6d, blue and red lines respectively). Each Monday, profile 2 makes the battery degrade faster than profile 1. This is because profile 2 contains seven partial cycles on Monday, while profile 1 represents only one. From Tuesday to Sunday, degradation rate is lower in profile 2 than in profile 1. After several weeks, it clearly appears that, despite of similar *SoC* levels, current rates and charge throughputs, profile 2 induces a lower average degradation rate than profile 1. Similarly, when comparing profiles 3 and 4, degradation is faster on Mondays for profile 3 respect to profile 4, but it is slower from Tuesday to Sunday. In a different way than for profiles 1 and 2, average degradation rates are similar in profiles 3 and 4 after a whole number of weeks (same cycle number): the acceleration that occurs every Monday on profile 3 is compensated by the deceleration of the rest of the week. As a conclusion, the degradation produced by combination of cycling and calendar periods can be different depending of the time distribution of charges and discharges (as for profile 2 respect to profile 1) or not (as for profile 3 respect to profile 4). The proposed model is able to reproduce a such particular behaviour inspired by experimental results, what could not be easily modelled by other approaches described in Section 2.

Other factors such as Δ*SoC* range, mean *SoC* or current rate and may influence $Q_F$. To explore how these factors can affect $Q_F$, other profiles were simulated (Table 4). From profiles in example 1, twelve supplementary profiles were generated by changing $SoC_{max}$, $SoC_{min}$ or current rate. Profiles 5 to 8 are designed to test the influence of Δ*SoC* range: $SoC_{min}$ is fixed to 0.6, then the weekly charge throughput is 2.8 *p.u.*

With profiles 9 to 12 we can test the influence of current rate by comparing the results from these profiles to those of profiles 1 to 4. Finally, profiles 13 to 16 are identical to profiles 1 to 4 but moving *SoC* levels (max, min, avg) 0.2 *p.u.* to the bottom. Rest times where adjusted to make average SoC ($SoC_{avg}$) correspond to 0.98, 0.82, 0.78 or 0.62, that is, $SoC_{max} - 0.02$ or $SoC_{min} + 0.02$.

In Figure 7 a selection of these results are shown. To explore how Δ*SoC* range influences $Q_F$, we can compare profiles 1 and 2 to profiles 5 and 6 respectively. $Q_F$ after 10 weeks under profile 5 was 26.51% of initial capacity compared to 19.62% under profile 1, that is, degradation rate was 35%

faster with $\Delta SoC$ equal to 0.4 *p.u.* When comparing the weekly profiles (profile 6 versus profile 2), the relative increase of the degradation rate was 39%. The influence of current rate is negligible, the difference between $Q_F$ under profiles 9 to 12 compared respectively to profiles 1 to 4 is lower than 1%. Finally, as shown by the results of simulations made with profiles 13 to 16, the influence of *SoC* level is very important, with degradation rates 15 to 40% slower than those of profiles 1 to 4.

These simulation results, which need to be validated by experiments, show that battery use management can have a significant influence on ageing. This model allows the evaluation of complex use scenarios, such as a fleet of vehicles: in some situations, it may be beneficial to use a vehicle intensively one day a week rather than using it regularly every day.

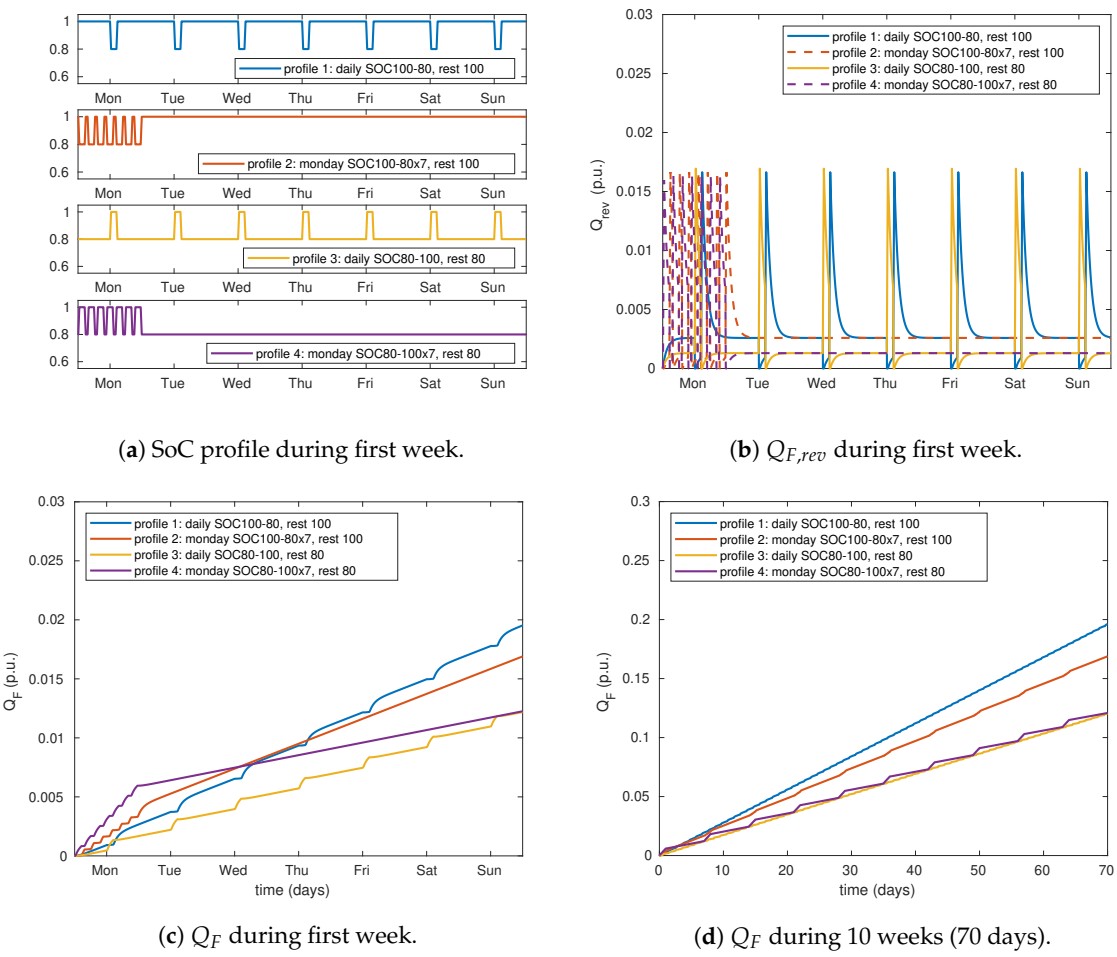

(**a**) SoC profile during first week.

(**b**) $Q_{F,rev}$ during first week.

(**c**) $Q_F$ during first week.

(**d**) $Q_F$ during 10 weeks (70 days).

**Figure 6.** Simulation results for cycling profiles between SoC 100 and 80% with current rates $C/2$. All profiles have the same weekly charge throughput: 1.4 *p.u.*, that is, seven times 0.2 *p.u.* Profiles 1 and 2 have the same average SoC (98%) as profiles 3 and 4 (82%).

**Table 4.** Simulation results for different use profiles. Underlined values are showing the differences between each four profile group (profiles 5 to 8, 9 to 12 and 13 to 16) respect to the original profiles (1 to 4).

| Profile Number | $SoC_{max}$ | $SoC_{min}$ | $SoC_{avg}$ | $I$ | Weekly Charge Throughput | $Q_F$ after 70 Days | Comments |
|---|---|---|---|---|---|---|---|
| | *p.u.* | *p.u.* | *p.u.* | *C* | *p.u.* | % | |
| 1 | | | 0.98 | | | 19.62 | daily SOC100-80, rest 100 |
| 2 | 1 | 0.8 | | 0.5 | 1.4 | 16.89 | monday SOC100-80x7, rest 100 |
| 3 | | | 0.82 | | | 12.03 | daily SOC80-100, rest 80 |
| 4 | | | | | | 12.08 | monday SOC80-100x7, rest 80 |
| 5 | | | 0.98 | | | 26.51 | daily SOC100-60, rest 100 |
| 6 | 1 | 0.6 | | 0.5 | 2.8 | 23.44 | monday SOC100-60x7, rest 100 |
| 7 | | | 0.62 | | | 11.31 | daily SOC60-100, rest 60 |
| 8 | | | | | | 11.35 | monday SOC60-100x7, rest 60 |
| 9 | | | 0.98 | | | 19.36 | daily SOC100-80 ($I = C/5$), rest 100 |
| 10 | 1 | 0.8 | | 0.2 | 1.4 | 16.54 | monday SOC100-80x7 ($I = C/5$), rest 100 |
| 11 | | | 0.82 | | | 11.64 | daily SOC80-100 ($I = C/5$), rest 80 |
| 12 | | | | | | 11.71 | monday SOC80-100x7 ($I = C/5$), rest 80 |
| 13 | | | 0.78 | | | 13.18 | daily SOC80-60, rest 80 |
| 14 | 0.8 | 0.6 | | 0.5 | 1.4 | 10.25 | monday SOC80-60x7, rest 80 |
| 15 | | | 0.62 | | | 10.17 | daily SOC60-80, rest 60 |
| 16 | | | | | | 10.12 | monday SOC60-80x7, rest 60 |

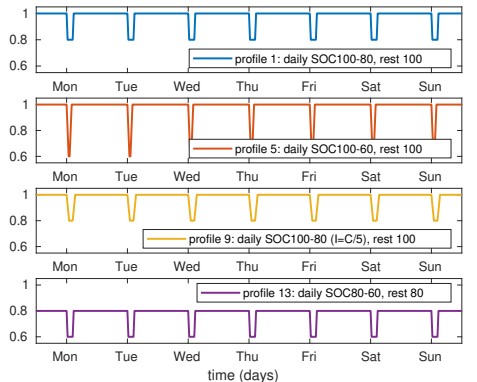

(**a**) Weekly SoC for profiles 1, 5, 9, 13.

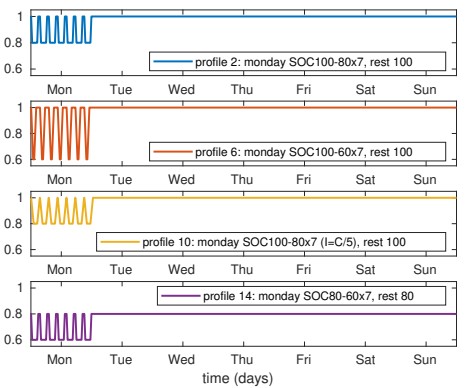

(**b**) Weekly SoC for profiles 2, 6, 10, 14.

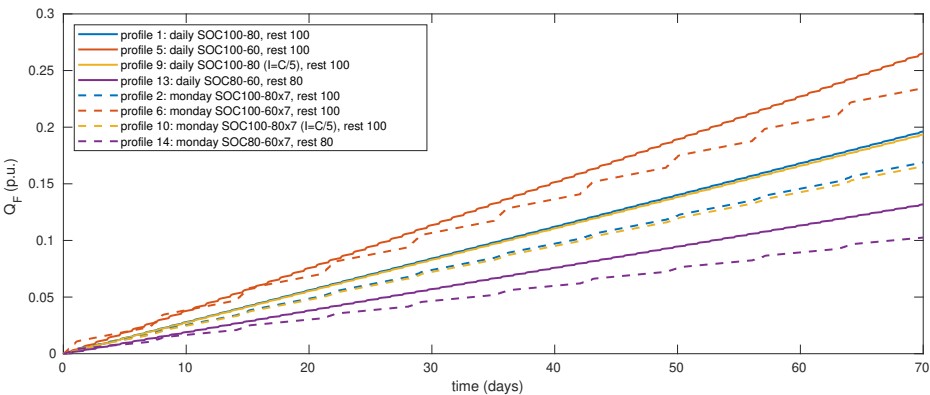

(**c**) $Q_F$ for profiles described in (a) (continuous lines) and (b) (dashed lines), same colours.

**Figure 7.** Simulation results for different *SoC* levels and current rates.

## 7. Conclusions

Battery ageing in electric vehicles is composed of calendar and cycling ageing. When cycling is performed at low current rates, typical cycling ageing mechanisms such as lithium plating or particle cracking can be neglected compared to the main calendar ageing mechanism in graphite based lithium-ion batteries: SEI growth. However, it has been found that cycling can influence subsequent calendar ageing. The combination between cycling and calendar ageing has a very non linear behaviour.

In this work we have modelled the capacity fade of NMC/C cells subjected to combined calendar and cycling ageing. The accelerated ageing test campaign was designed to investigate battery ageing at *SoC* levels and current profiles representative of electric vehicle's use.

The model identification consisted in two steps. In the first one, a calendar ageing model based on the Eyring law is proposed and a parameter identification is performed on calendar ageing experiment results. In the second step, cycling ageing experimental results are combined to the firstly identified calendar ageing model to obtain the combined ageing model parameters.

The proposed combined ageing model lies on the formulation of a two-step reaction rate. With the analogy between reaction rate equations and capacity fade, this ageing model is simple but effective: based on only two differential equations and seven parameters, it can reproduce the capacity evolution of lithium-ion cells subjected to cycling profiles similar to those found in electric vehicle applications. Moreover, the strong non-linearity of the cycling-calendar combination on ageing can be simulated with this model whereas it cannot be explained with models based on weighted Ah or event-oriented modelling approaches.

The obtained model opens the perspective of a wide range of applications, for example: battery use assessment, optimal electric vehicle charge scheduling or plug-in hybrid electric vehicle energy management strategy design.

Further work will consist in expand the domain of study. Especially, it will be important to study the combination of cycling and calendar ageing at *DoD* levels different than 20%, at *SoC* levels below 50% and at colder temperatures: this would allow to estimate other model parameters as, for instance, the activation energy ($E_a$).

Other studies could consist in thermal cycling, which may cause different ageing mechanisms to coexist, particularly lithium plating and SEI growth. The succession of ageing mechanisms of different nature may lead to multiple interactions between mechanisms. Analysing such interactions will be helpful to better understand battery degradation in real operation conditions.

**Author Contributions:** Conceptualization, Methodology and Investigation: E.R.-I., P.V. and S.P.; Project Administration, Data Curation, Software, Visualization and Writing Original Draft: E.R.-I.; Resources, Supervision and Writing Review: P.V. and S.P. All authors have read and agreed to the published version of the manuscript.

**Funding:** This research received no external funding.

**Conflicts of Interest:** The authors declare no conflict of interest.

## Abbreviations

The following abbreviations are used in this manuscript:

| | |
|---|---|
| LFP/C | Lithium Iron Phosphate/graphite |
| NMC/C | Nickel Manganese Cobalt/graphite |
| ODE | Ordinary Differential Equation |
| RPT | Reference Performance Tests |
| SEI | Solid Electrolyte Interface |
| SoC | State of Charge |

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
