# Peer review of "Modelling Lithium-Ion Battery Ageing in Electric Vehicle Applications—Calendar and Cycling Ageing Combination Effects"

_batteries, doi:10.3390/batteries6010014_

Round 1

Reviewer 1 Report

The authors tackle a very interesting topic that clearly will be an issue in the comming years.

They have done a good job in trying to mix the ageing behavior of cells in one single model.

The problem that I found is that the correlation found is not very visible in Figure 5. I suggest to separate calendar and cycling. As the colors are very similar, it is hard to see what experimental results correspond to the simulation results and if the simulation certainly follows any trend or is is a rough approximation.

Additionally, the authors are right in mentioning that ageing is affected by SOC, temperature and curent (I). However, they did not presented one important factor, the DoD, which is clearly and certainly an issue (please add references in that sence). This Depht of Discharge effect is even declared by the same cell manufacturers in many cases, so I really think the paper should mention it.

In fact, the tests do only run on a 100 to 80% SoC, which is not much (and cells do like small DoD).

Then, finally, I do find too much self-citations. It is true that some of them are important as they refer to the empiric results from previous publications. But, for instance, [13] is clearly for free and could be avoided. Please notice that there has been an extensive research in this field and you can cite other authors instead (this is a good way to put the relevance of the actual paper into the state of art).

Author Response

First of all, please allow us to thank the reviewer for the effort made in reviewing’s process of the article. We are convinced that thanks to the raised remarks, the quality of this article will be improved in a significant way. In the revised version, new included text is in blue. Hereby you will find the answers to your comments point by point:

- Concerning the figure 5: we have split the figure in two subfigures as you suggested (calendar and cycling).

- We agree with the reviewer that depth of discharge is an important factor of battery degradation and it is a real issue to determine in which conditions DoD can affect or not battery ageing and to quantify this degradation. In this way, the following new content (four bibliographic references) was included (lines 58 to 68):

The degradation of lithium-ion batteries is assumed to depend on three fundamental factors: T, I, and SoC. It should be underlined that not only the instantaneous value of these three factors, but also their temporal variations can impact the battery life. For example, in ageing tests campaigns in [6] and [7] batteries were cycled at different levels of SoC and different amplitudes (DoD). In these two works an important influence of the cycling amplitude was found. As pointed by [5], SoC (or DoD) influence on ageing is not simple to analyse, because in some situations low SoC levels (high DoD) can be beneficial while it could be harmful in other cases. Finally, a very non linear SoC dependence of cycling ageing was identified experimentally by [8]. In that work, five cycling tests of the same SoC amplitude (20%) at different average SoC levels were carried out. Cycling at very low and at very high levels of SoC (0 to 20% and 80 to 100%) caused respectively the slowest and the fastest degradations, but no big difference was found between intermediary SoC levels (20 to 40%, 40 to 60% and 60 to 80%).”

This work is based on results from the experimental campaign described in [3]. As you may know, battery ageing tests are very heavy to perform in question of time, material and human availability. These tests will be certainly completed by other containing different SoC profile shapes, lower levels of average SoC and different SoC amplitudes. In this way, some new text is also included in the conclusion (lines 392, 393):

Further work will consist in expand the domain of study. Especially, it will be important to

study the combination of cycling and calendar ageing at DoD levels different than 20%...”

- Self citation in line 103 (calendar ageing models) was replaced by two other citations (Grolleau et al., Eddahech et al.).

Reviewer 2 Report

This is a very interesting study on the battery degradation during operation and the rest status. I would recommend to accept the manuscript in the current format, while I can suggest the authors to keep investigating more complicated working conditions in their future works such as the degradation modelling for cells with SOC below 50 or even below 20% where internal resistance might increase as well.  

Author Response

First of all, we would like to thank the reviewer for their effort in reviewing this article. We completely agree with the reviewer that it will be interesting to study battery degradation at other levels of stress, especially lower SoC levels. Some new content is included to the conclusion in that way.

Round 2

Reviewer 1 Report

The authors did what I asked in most of my questions. However, what I asked in one case in particular had an intrinsecal  second step that the authors did not perform, I asked to separate fig.5 in two because it was hard for me to identify if the simulation really followed the real experimental measurements. The authors effectively separated fig.5 in two different parts.

This was useful to me to really see that the model do fits somehow to the calendar ageing although I would tend to believe (according to different literature) that they should follow a non linear curve, contrarily to what it is presented (see how green and cyan do not fit to linear trends).

Then, Fig.5 b confirms my previous doubts. The model does not really achieves a good fitting for cicling. For instance:

Profile b shows errors of capacity fade of twice what was measured (simulating minus degradation than real measurements). Profile c and d have no measurements to compare with. Profile e has not enough measurements to validate the results (and seems that simulations give worse results than what really occurs).

Seeing that, I do not thing the model is robust enough and that it should be validated in more depth (either more tests, either more time).

Notice that the authors had even presented a new paragraph after fig.2 where they say:

"A result that may seem surprising appears when comparing the capacity fade evolution of cells subjected to profiles a and b. These two profiles are identical, except for the second partial discharge of the profile a (Figure 1). All ageing factors (T, I, average SoC) are similar between these two profiles, except for the depth of discharge: DoD in profile a is twice compared to DoD in profile b. However, the cells subjected to profiles a or b evolved in the same way despite the fact that the DoD is very different (Figure 2)."

Even if writting this statement, they maintain the model simulation results that indicate that following profile a battery ages 20% faster than following profile b. How do the authors can justify this?

As a consequence, the results have no real application to me. Why should I take the results of the simulation done afterwards as good ones if the simulation is not correlated to the measurements?

Maybe I'm wrong, and, if that is so, I suggest the authors to indice me the reasons they cosnider to present the model as a valuable tool.

Since that is done, I maintain my major revisions status (the paper should not be published as is).

Round 3

Reviewer 1 Report

OK,

Everything is answered and clear. I hope future works led you achieve the accuracy missing in this case. Good job

Thank you.

Lluc